# Clinical and Non-Clinical Determinants of the Effect of Mechanical Thrombectomy and Post-Stroke Functional Status of Patients in Short and Long-Term Follow-Up

**DOI:** 10.3390/jcm10215084

**Published:** 2021-10-29

**Authors:** Anetta Lasek-Bal, Łukasz Binek, Amadeusz Żak, Sebastian Student, Aleksandra Krzan, Przemysław Puz, Wiesław Bal, Urszula Uchwat

**Affiliations:** 1Department of Neurology, School of Health Sciences, Medical University of Silesia, 40-055 Katowice, Poland; aurii9@o2.pl (A.Ż.); aleks.krzan@gmail.com (A.K.); ppuz@o2.pl (P.P.); 2Department of Neurology, Upper-Silesian Medical Centre, Medical University of Silesia, 40-055 Katowice, Poland; bunchet1@gmail.com (Ł.B.); uuchwat@gmail.com (U.U.); 3Faculty of Automatic Control, Electronics and Computer Science, Silesian University of Technology, 44-100 Gliwice, Poland; sebastian.student@polsl.pl; 4Biotechnology Center, Silesian University of Technology, 44-100 Gliwice, Poland; 5Department of Outpatient Chemotherapy, Maria Skłodowska-Curie Memorial Cancer Centre and Institute of Oncology, 44-101 Gliwice, Poland; balwieslaw@gmail.com

**Keywords:** stroke, thrombectomy, functional state

## Abstract

To date, inconsistent results evaluating the effect of parameters on mechanical thrombectomy (MT) outcomes in stroke-patients have been published. This study aimed to identify the key parameters for functional status after MT in stroke-patients in short and long-term follow-up. Method: The study analysis focused on the relevance of selected clinical and non-clinical parameters to the functional status of the patients after MT. Results: 417 stroke-patients (mean age 67.8 ± 13.2 years) were qualified. Atrial fibrillation, and leukocytosis were significant for the neurological status on the first day of stroke (*p* = 0.036, and *p* = 0.0004, respectively). The parameters with the strongest effect on the functional status on day 10 were: age (*p* = 0.009), NIHSS (*p* = 0.002), hyperglycemia (*p* = 0.009), the result in TICI (*p* = 0.046), and first pass effect (*p* = 0.043). The parameters with the strongest effect on the functional status on day 365 were: age and NIHSS on the first day of stroke (*p* = 0.0002 and 0.002, respectively). Leukocytosis and the neurological status at baseline were key parameters associated with ICB after MT (*p* = 0.007 and *p* = 0.003, respectively). Conclusions: Age and neurological status in the ultra-acute phase of stroke are crucial for the functional status in short and long-term observations of patients treated with mechanical thrombectomy. Atrial fibrillation, hyperglycemia, and inflammatory state are relevant to the short-term post-stroke functional status. First pass effect and the degree of post-interventional reperfusion are important technical parameters to the short-term functional status. Neurological status and white blood count during the acute phase are associated with a high rate of post-procedural intracranial bleeding.

## 1. Introduction

In 2013, after the results of three studies on the efficacy of mechanical thrombectomy (MT) in acute stroke were analyzed, the future of MT was uncertain [1,2,3]. Randomized trials published two years later initiated a new era in endovascular therapy for stroke in the course of acute large vessel occlusion (LVO) [4,5,6,7,8]. The HERMES meta-analysis, which collected data from over 1200 patients, has provided conclusive evidence for both the safety and efficacy of MT [9]. It was demonstrated that 46% of patients undergoing MT in the course of acute stroke could function independently three months later (as opposed to 26.5% of individuals in the pharmacological treatment group). Those who also benefited from endovascular treatment (OR 3.68, 95% CI 1.95–6.92) were also patients aged ≥80 as well as randomized patients >300 min after the onset (1.76, 1.05–2.97). Recombinant tissue plasminogen activator (rt-PA) was previously used in 83% of patients undergoing MT. However, patients not eligible for rt-PA also gained some distinct clinical benefits from MT (2.43, 1.30–4.55). The 90-day mortality rate and the incidence of symptomatic intracranial bleeding (ICB) in the interventional treatment group were not significantly different than those observed following rt-PA. Further studies and meta-analyses stress the importance of MT in the treatment of stroke by indicating an ever-increasing potential associated with the length of the therapeutic window [10,11]. Although MT results in recanalization of more than 80% of the arteries involved in the intervention, no more than 50% of patients manage to return to complete independence within three months following stroke [9]. This means that in half of the patients, a successful angiographic outcome does not guarantee optimal clinical benefits. Therefore, attempts are being made to identify the parameters to determine the outcome of endovascular therapy in acute stroke patients. The most common findings are age, the patients’ neurological status during the ultra-acute stage of stroke, and the ASPECTS score [12,13,14]. Atrial fibrillation, diabetes, previous exposure to anticoagulation, the morphotic parameters of blood, and inflammatory parameters all have a conflicting status in several studies [12,15,16,17]. If factors affecting the safety and efficacy of MT could be identified, that results would influence the selection of patients for the procedure and thus optimize the efficacy of endovascular procedures in stroke.

This study aimed to identify the clinical and non-clinical prognostic parameters for functional status in stroke patients undergoing MT in both short and long-term follow-up.

## 2. Materials and Methods

The retrospective study included patients who underwent MT in the acute period of stroke at the Upper-Silesian Medical Centre of the Silesian Medical University in Katowice over two years (2019–2020). All patients included in the study were analyzed for:
their age at the time of their first-ever stroke;their white blood cell counts (WBC), C-reactive protein (CRP), platelet (PLT), hemoglobin (Hb), glucose and troponin (TnI) concentration on the first day of hospitalization (at admission);their neurological status on the first day (at baseline, before MT) of stroke evaluated on the NIHSS (National Institute of Health Stroke Scale);comorbidities, such as atrial fibrillation (AF), arterial hypertension (AH), coronary heart disease (CHD), diabetes mellitus (DM), lipid disorders (LD), and >70% atherosclerotic carotid artery stenosis (CAS, ipsilaterally to stroke);exposition to antithrombotic/anticoagulant therapy before stroke;echocardiographic parameters (left atrium size, ejection fraction);angiographic effect of MT (according to TICI)beneficial angiographic effect (TICI 2b-3) after the first pass (FPE, first pass effect);their functional status according to modified Rankin scale (mRS) on days 10, 30 and 365 following stroke.

In the process of causative diagnostics for stroke, we tried to find the vascular risk factors for stroke in each patient. We initially managed to identify the potential risk factors for stroke based on patient history of cardiac disease (atrial fibrillation, ischemic heart disease, past myocardial infarction, etc.), arterial hypertension, diabetes mellitus, and lipid disorders. Regardless of the results of the neck/head CT angiography performed before MT, each patient within 3 days after admission had a Doppler ultrasound performed to assess the status of carotid/vertebral arteries and the hemodynamic parameters. Similarly, each patient had a transthoracic echocardiogram performed. Those below 60 years of age underwent a transesophageal echocardiogram. Each patient had their baseline tests performed: blood count, ionogram, renal and hepatic parameters, lipid profile, and glucose test. Patients below the age of 60 had an additional blood test performed to check for coagulation disorders (proteins C and S levels, antithrombin concentration, factor VIII activity test, factor V Leiden mutation molecular tests, prothrombin 20,210 mutation tests), antiphospholipid syndrome, vasculitis, infections, and connective tissue diseases. In each patient with no AF in the basic electrocardiogram recording, a 24-h electrocardiography device monitoring was performed within 14 days; and the 30-day electrocardiogram was obtained by telemetry if no AF was recorded in the 24-h test. AF was diagnosed based on previous patient records or ECG, or 24-h ECG monitoring performed during stroke-related hospitalization (≤14 days after onset) or 30-day ECG telemonitoring.

Lipid disorders were diagnosed when at least 1 of the following features was present: hypercholesterolemia with total cholesterol concentration ≥190 mg/dL (≥5 mmol/mL) or LDL-C ≥115 mg/dL (≥3 mmol/L) and/or hypertriglyceridemia with triglyceride concentration ≥150 mg/dL (≥1.7 mmol/L).

The degrees of stenosis in the common carotid artery and/or internal carotid artery were assessed according to the NASCET criteria. Hemorrhagic lesions were evaluated based on head computed tomography (CT) performed at 24 h after MT and the ECASS scale. Intracranial bleeding (ICB) was defined as symptomatic when the neurological status deteriorated due to ICB, with an increase of at least 4 points on the NIHSS during 24 h following MT. The correct diameter of the left atrium (LA) was considered to be ≤40 mm, and the correct fraction ejection (EF) value > 50%. The patients with hemoglobin < 10 g/dL as well as patients with platelet < 100 × 109/L) were not qualified to our study.

The effect of selected parameters on the patients’ functional status after MT was analyzed. These parameters were divided into clinical parameters, i.e., disease-related parameters (AF, AH, CHD, DM, LD, >70% CAS) or those related to the patients’ clinical status, i.e., the neurological status on the first day of stroke, and other selected non-clinical parameters. Those selected non-clinical parameters were divided into the following groups: demographic parameters (age, sex), blood count (WBC, Hb, PLT), biochemical parameters (CRP, glucose, and TnI), echocardiographic parameters (left atrial, ejection fraction), therapy-related parameters (pre-stroke anticoagulation, pre-stroke antiplatelet therapy, rt-PA iv, hemicraniectomy), lifestyle-related parameters (nicotinism), and radiological parameters (TICI, ICB).

### 2.1. Description of the Mechanical Thrombectomy Method

Mechanical thrombectomy procedures are performed at the Interventional Neuroradiology department equipped with a state-of-the-art device: the Siemens Axiom Artis Zee Bi-Plane (Siemens, Munich, Germany). If the angiosuite was not available, the procedure could also be performed at the Vascular Surgery Cath-lab or Interventional Cardiology Cath-lab equipped with appropriate monoplane machines. Based on personal preferences and the clinical condition of the patient, general anesthesia or conscious sedation was used. Patient qualification, logistics, and postoperative care was provided by neurologists.

Most mechanical thrombectomy procedures were performed with the use of stent retrievers (Soliter, Trevo, Catch, Tiger) and an 8F Balloon Guide Catheter (Flow Gate 2). For the remaining procedures, distal aspiration (Penumbra, Cathalist, Sofia) or combined stent retrievers and distal aspiration were used.

The presented study was not a medical experiment and as such it was not required to be evaluated by the Bioethics Committee of the Medical University of Silesia in Katowice.

### 2.2. Statistical Analysis

The patient group was analyzed for age and both clinical (as listed above) and non-clinical factors (as listed above) relevant to the functional status at three time-points following MT: day 10th after stroke onset, 30th and day 365th following the onset of stroke.

Regression analysis was performed to assess the effect of the following parameters on the patients’ neurological state (NIHSS at baseline): clinical parameters (LD, AF, AH, DM, CAS, CHD) and non-clinical parameters (age, gender, CRP, WBC, TnI, EF, LA, nicotinism).

Next, the clinical parameters (LD, AF, AH, DM, CAS, CHD, NIHSS at baseline) and non-clinical parameters (age, gender, CRP, WBC, TnI, rt-PA iv, FPE, nicotinism) were analyzed to identify these important parameters for obtaining the 2b-3 TICI score after MT.

Further, a multifactorial analysis was carried out to identify independent factors for ICB in CT of the head 24 h after MT. Clinical parameters (LD, AF, AH, DM, CAS, CHD, NIHSS) and non-clinical parameters were analyzed (age, gender, CRP, WBC, PLT, TnI, rt-PA iv, TICI, nicotinism)

Regression analysis was performed to assess the effect of the following parameters on the patients’ functional status on day 10: clinical parameters (LD, AF, AH, DM, CAS, CHD, NIHSS) and non-clinical (age, gender, CRP, WBC, TnI, glucose, EF, LA, rt-PA iv, TICI, ICB, FPE, hemicraniectomy, nicotinism).

Next, regression analysis was performed to assess the effect of the following parameters on the patients’ functional status on day 30: clinical parameters (LD, AF, AH, DM, CAS, CHD, NIHSS) and non-clinical parameters (age, gender, CRP, WBC, glucose, TnI, EF, LA, rt-PA iv, TICI, ICB, hemicraniectomy, nicotinism).

Next, regression analysis was performed to assess the effect of the following parameters on the patients’ functional status on day 365: clinical parameters (LD, AF, AH, DM, CAS, CHD, NIHSS) and non-clinical parameters (age, gender, TICI, ICB, hemicraniectomy, nicotinism).

Multivariable models were built by using ordinal logistic regression for ordinal outcomes. The model variable selection procedures included automatic parameter selection (stepwise, forwards and backwards) based on the the Akaike Information Criteria (AIC) and Bayesian information criterion (BIC) criteria. The reduced model (for mRS on days 30th and 365th) included only the best parameters and the highest AIC and BIC criteria. To evaluate the accuracy of model predictions, the “leave-one-out” procedure was used to avoid data leakage which can cause overfitting and the multiclass AUC estimator were used. All statistical analyses were performed using R version 3.6.1.

## 3. Results

A total of 417 stroke patients (mean age 67.8 ± 13.2 years; mean 70 (20–92)); female: 47.7%) hospitalized between 2019 and 2020, who were treated with mechanical thrombectomy due to stroke were qualified for the retrospective study. The subject group consisted of 30.17% of all the patients treated for ischemic stroke at our Stroke Centre in the above-mentioned period (1382 stroke patients). Among all of the patients who were included in the study, 66% were treated according to the logistic model drip-end-ship, 61% received rtPA. The strokes etiology according to the TOAST classification presented as follows: large artery atherosclerosis-in 179 (42.92%) patients, cardioembolic in 108 (25.89%), other determined in 15 (3.59%) and undetermined in 115 (27.57%) patients.

Less than 50% of patients were exposed (at least one month) to antithrombotic therapy -antiplatelet (aspirin was used by 115 patients, clopidogrel by 19 patients) or anticoagulant (24 patients were using warfarin, 16 were using rivaroxaban, eight were using dabigatran, and seven were using apixaban). The correct therapeutic results of INR (2.5–3.2) were observed in 16 patients (66.6% of all patients were using warfarin).

ICB was observed in 95 patients in control CT of the head 24 h after MT (according ECASS: HI 1 in 32 patients, HI 2 in 29, PH 1 in eight, PH 2 in 16, PH r1 in six, and PH r2 in four). Neurological deterioration due to ICB (increase of at least four points in NIHSS during the 24 h following MT) was observed in 21 patients.

The demographic and clinical data are listed in Table 1.

Out of all the clinical factors, only atrial fibrillation was significant (*p* = 0.036) for the neurological status (at baseline, according to NIHSS) on the first day of stroke. Leukocytosis on the first day of stroke was the only one significant non-clinical factor (*p* = 0.0004).

The key parameter for obtaining TICI 2b–3 could not be found. A trend suggesting the significance of age for the angiographic effect was observed. However, it had no statistical significance (*p* = 0.057).

An abnormal WBC count and the neurological status at baseline (NIHSS) in the ultra-acute phase of stroke were key parameters associated with ICB on head CT at 24 h following MT (*p* = 0.007 and *p* = 0.003, respectively).

Hyperglycemia ≥160 mg/dL upon hospital admission, had a negative effect on achieving a good functional status (0–2 points on the mRS) on day 10th of disease (OR 0.598, 95% CI 0.373–0.795, *p* = 0.009). Similarly, the technical parameters: the recanalization according TICI and first pass effect (FPE) were significantly important for the functional status on the 10th day after MT (OR 0.506, 95% CI 0.255–0.976, *p* = 0.046 and OR 0.5495, 95% CI 0.261–0.898, *p* = 0.043, respectively). The model based on attributes TICI (OR 0.489, 95% CI 0.262–0.899, *p* = 0.043) and age (OR 1.034, 95% CI 0.009–1.06, *p* = 0.009), NIHSS (OR 1.103, 95% CI 1.039–1.174, *p* = 0.002), and DM (OR 1.827, 95% CI 0.884–3.93, *p* = 0.112) showed good predictive attributes for the functional status of patients in the acute period (on day 10 after onset).

In multivariate analysis, the parameters with the strongest effect on the functional status (mRS) at day 30 were age and NIHSS at baseline (Table 2 and Table 3).

The parameters with the strongest effect on the functional status on day 365 were age and NIHSS on the first day of stroke (Table 4 and Table 5).

## 4. Discussion

The main result of this study is a conclusion that age and the neurological status in the ultra-acute phase of stroke are the key parameters influencing the post-stroke functional status in both short and long-term follow-up in patients undergoing MT. Comorbid atrial fibrillation, diabetes and lipid disorders worsen the prognosis related to the functional status in the acute phase of stroke in this group of patients. The results also indicate an adverse effect of elevated leukocytosis on the first day of stroke, and of elevated CRP, on the risk of ICB and the functional status of patients on the first days following onset. A positive angiographic outcome and the effect of the first pass increased the chances of improvements in the functional status during the first week following the onset of stroke.

To date, several independent factors have been identified by some authors which reportedly worsen the functional status of patients undergoing MT for stroke. These include older age, low ASPECTS scores, and high NIHSS scores [12,13,14,17,18]. In terms of the importance of age and neurological status, we have confirmed the reports of other authors; we have also gained more experience concerning the influence of a patient’s clinical profile on the outcome of endovascular therapy and their post-stroke functional status in the acute period of stroke. The ability to identify the non-radiological parameters affecting the outcome of MT can be of significant importance in patients with stroke present for at least 6 h or in those whose stroke duration is unknown [19]. No significant importance regarding thrombolytic therapy preceding MT was indicated in our study.

To date, inconsistent results evaluating the effect of DM and AF on MT outcomes have been published [13,15]. AF is a common cause of LVO that necessitates endovascular treatment [9,20,21,22]. MR CLEAN sub-analysis failed to confirm a significant effect of AF on MT outcomes [23,24]. Several separate retrospective studies found a negative effect of AF on the functional status of patients treated with endovascular therapy for stroke [12,15,16]. In contrast, the STAR registry, which accumulates the data from more than 4000 patients undergoing MT, did not find any negative effect of AF on MT outcomes [12]. Among our patients, AF did not increase the risk of intracranial hemorrhagic complications. However, Huang et al. (2021) showed a significant increase of ICB in AF patients after MT compared to those with no AF. This did not significantly influence their functional status or mortality at three months after stroke [25]. In general, MT is considered to be safe in terms of the risk of increased ICB in patients given anticoagulant treatment [26,27,28,29]. However, there are studies which indicate that AF increases the risk of ICB even in patients with low INR [25]. The presented study also evaluated the potential influence of prior anticoagulant therapy; it did not show any relationship with the parameters. Moreover, according to current published results, the initiation of anticoagulation in secondary stroke prevention does not increase the risk of ICB in patients treated with MT [30]. This is an important observation regarding the safety of MT in patients with LVO stroke related to AF. In line with the present multicenter experience, 25–50% of patients undergoing MT suffer from AF and as much as half of these have been using anticoagulation treatment in the period of stroke onset [12,20,22,23,25].

More and more reports have been published to confirm the negative effect of diabetes on the outcome of MT. However, the results are quite inconsistent [17,31]. Diabetes mellitus is not only the risk factor for atherosclerosis and stroke, but also for stroke recurrence and an early neurological deterioration in acute stroke patients [32,33,34]. DM also increases the risk of hemorrhagic transformation of cerebral infarction in patients undergoing thrombolysis and MT [35,36]. According to the analysis of the SWIFT study results, hyperglycemia increases the risk of a worse outcome at three months in stroke patients [37]. In the presented study, we demonstrated an association between DM and a worse functional status in the acute period of stroke. A hyperglycemic environment is deemed to promote a poorer ability to develop collaterals and an increased volume of the infarction focus [38]. Insufficient collaterals result from diabetic microangiopathy. It has been shown that the condition of collaterals is associated with hyperglycemia in stroke patients [39]. The condition can impair revascularization and reperfusion and increase reperfusion injury to the neural tissue [38]. Regardless of the condition of collaterals in patients with DM, metabolic changes in DM can impair the preservation of the penumbra. While diabetes is a recognized risk factor for cerebrovascular diseases, the consequences of stress hyperglycemia are not well-established. The latter is usually defined as spontaneously resolving hyperglycemia after acute illness dissipation. Stress hyperglycemia is mediated by pro-inflammatory cytokines that cause a stress response with excessive gluconeogenesis, glycogenolysis, and insulin resistance. [40]. Some studies have reported that stress hyperglycemia increases the risk of poor outcomes in stroke patients [41,42,43].

The results of the presented study exhibit a correlation between the parameters of the inflammatory condition and functional status during the acute stroke period. As an inflammatory process, coagulation disorders and atherosclerosis all play key roles in the pathogenesis of acute cerebral ischemia. Activated platelets adhere to ruptured parietal atherosclerotic plaques and damage vascular walls while releasing soluble CD40 ligand and β-globulin which stimulate lymphocytes and other inflammation factors, thus exacerbating damage to the neural tissue [44,45,46]. IL-6 is a marker of futile reperfusion in the setting of MT [47]. In our previous studies, an increase in WBC counts and CRP levels was associated with a worse course in the acute phase of stroke and post-stroke functional status [48]. We also demonstrated that chronic paranasal sinusitis exerts an unfavorable effect on the status of patients who underwent MT for stroke [49]. The inflammatory process probably stimulates the development of stroke-inducing atherosclerosis. In these cases, thrombus contains more fibrin than red blood cells, which makes it potentially more difficult to be removed (which is not the case with cardiogenic thrombus) [50,51].

The relevance of the technical aspects of the intervention to its clinical efficacy was also evaluated in the presented study. We showed that the effect of the first pass is important for the functional status in the first days of the disease. According to the current multicenter experience, the proportion of first pass effect ranges from 19% to almost 60% [52]. It appears that efficacious time management (door-to-puncture and puncture-to-reperfusion) increases the efficacy of the procedure. Recanalization after the first passage has been shown to increase the success of MT, also by the other authors [52,53,54,55]. Identifying the factors which influence the first-pass effect may help to optimally select the patients for MT based on pre-interventional risk factors and their possible modification options. According to the experience of other authors, the FPE effect depends on age, sex, type of sedation, and the use of a balloon guide catheter [52]. The profiling of patients in a clinical and non-clinical context may be important due to the prolongation of the therapeutic window for thrombectomy [11].

Knowing the relevance of separate clinical and non-clinical parameters to MT outcomes will support a better selection of patients based on individual factors and will optimize therapy in terms of both efficacy and safety. Further studies in this respect and the exchange of experiences between the centers offering endovascular therapy in stroke are required.

### Study Limitations

Our study has some limitations, the most important being: a single-center study, a retrospective study type, and the lack of an ASPECTS analysis before MT. The latter limitation was due to incomplete data. The majority of patients were transferred to our centre from other hospitals (in the drip-and-ship model) and we did not perform the CT of the head before MT.

## 5. Conclusions

The age and neurological status in the ultra-acute phase of stroke are crucial for the functional status in short and long-term observations of patients treated with mechanical thrombectomy.

Atrial fibrillation, hyperglycemia and inflammatory state are relevant to the short-term post-stroke functional status.

First pass effect and the degree of post-interventional reperfusion are important technical parameters to the short-term functional status.

Neurological status and white blood count during the acute phase are associated with a high rate of post-procedural intracranial bleeding.

## Figures and Tables

**Table 1 jcm-10-05084-t001:** Characteristics of the study group.

Age Mean ± SD; Med. (Range)	67.8 ± 13.2; 70 (20–92)
Female	199 (47.7%)
AF	108 (26.0%)
CHD (data for *n* = 408)	221 (52.9%)
DM	105 (25.3%)
LD	168 (40.4%)
AH	315 (75.5%)
CAS (ipsilateral ICA ≥ 50%)	51 (12.1%)
Nicotinism (data for *n* = 310)	115 (37.1%)
Antiplatelet therapy before MT	134 (32.1%)
Anticoagulation therapy before MT	55 (13.2%)
rt-PA iv	254 (61.2%)
Thrombocytopenia (data for *n* = 412)	27 (6.6%)
Leukocytosis	95 (22.7%)
Anemia	33 (7.91%)
High CRP levels	118 (28.2%)
High troponin concentration (data for *n* = 374)	202 (54.0%)
Admission hyperglycemia (≥160 mg%)	91 (21.82%)
INR out of therapeutic range (in treated patients)	39 (70.9%)
MT in drip-and-ship model	274 (65.7%)
Time stroke onset- first head CT, mean (range)	198 ± 7.8 min (119–315)
Time stroke onset- groin puncture, mean (range)	242 ± 12.3 min (80–368)
Time of MT duration, mean (range)	111 min (24–199)
FPE	176 (42.2%)
NIHSS; mean ± SD; med. (range); IQR (range Q1 Q3)	12.8 ± 5.6; 12 (0–43); 7 (9–16)
TICI 2b–3	274 (65.7%)
ICB; sICB	95 (22.7%); 21 (5%)
mRS 10 d; mean ± SD; med. (range); IQR (range Q1 Q3)	3.87 ± 1.75; 4 (0–6); 2 (3–5)
mRS 30d; mean ± SD; med. (range); IQR (range Q1 Q3)	3.32 ± 1.69; 3 (2–5); 4 (0–6)
mRS 12m; mean ± SD; med. (range); IQR (range Q1 Q3)	3.14 ± 1.52; 3 (0–6); 3 (2–5)

AF—atrial fibrillation, AH—arterial hypertension, CAS—carotid artery stenosis, CHD—coronary heart disease, DM—diabetes mellitus, FPE—first pass effect, ICB—intracranial bleeding, INR—international normalized ratio, LD—lipid disorders, MT—mechanical thrombectomy, mRS—modified Rankin Scale, mRS 10d-mRS on 10th day of stroke, mRS 30d-mRS on the 30th day of stroke, mRS 12m-mRS 1 year after stroke onset, NIHSS-NIHSS on the 1st day of stroke, PLT—platelet, sICB—symptomatic intracranial bleeding, TICI—thrombolysis in cerebral infarction, rt-PA iv—intravenous recombinant tissue plasminogen activator.

**Table 2 jcm-10-05084-t002:** Ordinal regression analysis of the influence of clinical and non-clinical parameters on mRS day 30.

Coefficients	OR	CI 95%	*p*-Value
Age	1.103	1.044–1.174	0.001
Gender	0.484	0.126–1.731	0.274
LD	0.554	0.265–1.126	0.108
AF	0.365	0.064–1.919	0.239
AH	0.743	0.053–1.828	0.238
DM	0.484	0.083–2.749	0.409
CAS	0.698	0.114–3.858	0.683
CHD	1.625	0.684–3.721	0.109
Nicotinism	0.373	0.115–1.62	0.163
CRP	1.827	0.884–3.931	0.111
WBC	0.294	0.08–1.002	0.05
TnI	0.554	0.265–1.126	0.101
EF	0.663	0.376–1.237	0.219
LAE	0.371	0.071–2.632	0.308
NIHSS	1.291	1.129–1.511	0.001
rt-PA iv	0.947	0.195–4.539	0.945
TICI	0.184	0.044–01.688	0.055
ICB	1.128	0.354–3.598	0.837
Hemicraniectomy	0.657	0.103–2.771	0.585

AF—atrial fibrillation, AH—arterial hypertension, CAS—carotid artery stenosis, CHD—coronary heart disease, CRP—C-reactive protein, DM—diabetes mellitus, EF—Ejection fraction, ICB—intracranial bleeding, LAE—left atrium enlargement, LD—lipid disorders, NIHSS—National Institutes of Health Stroke Scale on the 1st day of stroke; rt-PA iv—intravenous recombinant tissue plasminogen activator, TICI—thrombolysis in cerebral infarction, TnI—troponin I, WBC—white blood cell.

**Table 3 jcm-10-05084-t003:** Ordinal regression analysis of the influence of selected parameters on mRS day 30 after model selection mcAUC = 0.706.

Coefficients	OR	CI 95%	*p*-Value
Age	1.048	(1.015–1.086)	0.006
NIHSS	1.166	(1.077–1.275)	0
LD	0.554	(0.265–1.126)	0.108

LD—lipid disorders, NIHSS—National Institutes of Health Stroke Scale on the 1st day of stroke.

**Table 4 jcm-10-05084-t004:** Ordinal regression analysis of the influence of clinical and non-clinical parameters on mRS day 365.

Coefficients	OR	CI 95%	*p*-Value
Age	1.098	1.038–1.171	0.002
Gender	0.77	0.197–2.858	0.699
NIHSS	1.269	1.107–1.491	0.002
rt-PA iv	1.412	0.29–7.004	0.668
DM	0.927	0.156–5.593	0.933
AH	1.099	0.265–4.716	0.897
CHD	0.89	0.186–2.653	0.575
AF	1.05	0.176–6.529	0.957
LD	0.742	0.199–2.713	0.651
CAS	0.434	0.051–2.887	0.408
Nicotinism	0.262	0.114–1.522	0.163
TICI	0.259	0.062–0.988	0.053
ICB	1.26	0.375–4.254	0.707

AF—atrial fibrillation, AH—arterial hypertension, CAS—carotid artery stenosis, CHD—coronary heart disease, DM—diabetes mellitus, ICB—intracranial bleeding, LD—lipid disorders, NIHSS—National Institutes of Health Stroke Scale on the 1st day of stroke; rt-PA iv—intravenous recombinant tissue plasminogen activator, TICI—thrombolysis in cerebral infarction.

**Table 5 jcm-10-05084-t005:** Ordinal regression analysis of the influence of selected parameters on mRS day 365 after model selection mcAUC = 0.704.

Coefficients	OR	CI 95%	*p*-Value
Age	1.098	(1.038–1.171)	0.0002
NIHSS	1.269	(1.107–1.495)	0.002
TICI	0.319	(0.086–1.097)	0.076

NIHSS—National Institutes of Health Stroke Scale on the 1st day of stroke, TICI—thrombolysis in cerebral infarction.

## Data Availability

Not applicable.

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
