# Peer review of "Clinical and Non-Clinical Determinants of the Effect of Mechanical Thrombectomy and Post-Stroke Functional Status of Patients in Short and Long-Term Follow-Up"

_jcm, 2021, doi:10.3390/jcm10215084_

Round 1

Reviewer 1 Report

This is an interesting study based on relatively huge cohort and concerning an important clinical topic. Please find below my comments, which could probably help The Authors to improve the manuscript:

Ad. Methods:

  • Clear statement what are the study aims would improve data interpretation. First, authors declare that: … study aimed to identify the key parameters for efficacy of MT in stroke-patients in short and long-term follow-up treated with MT. However, further they state that … the key parameter for obtaining TICI 2b–3 could not be identified. In results, they generally present and conclude factors for the functional status in short and long-term observations. Thus, regarding presented material, in my opinion authors could simply declare, that they aimed to assess factors for functional outcome after MT in stroke patients.
  • Authors declare that they analyzed selected clinical and non-clinical factors, however they do not present selection criteria. Thus, Authors should present selection criteria, to avoid suspicion for “salami slicing” policy.
  • In my opinion, too little is told about the selection criteria for variables involved into regression models. Description: …. To evaluate the accuracy of model predictions, the “leave-one-out” procedure and the multiclass AUC estimator were used…. is probably too sparse. AUC results are neither presented in the results.

Ad. Results  

Some imprecisions and unclarities exist, that should be modified for better quality of the manuscript.

  • First, both in text and table 1 & 2 authors present some parameters, being described as … key parameters (in text - … An abnormal WBC count and the neurological status (NIHSS) in the ultra-acute phase of stroke were key parameters associated with ICB) or as those with the strongest effect (Tables). It remains unclear if The Authors mean that other variables has no statistically significant effect or impacted less? However, both other covariables and the significance of its impact should be presented or authors should prove why they have not been involved into the regression models.   
  • Term lipid disorders(Table 3) is too general and should be clarified.

Ad. Discussion

Some contradictions exist, that should be avoided, as follows:

  • The Authors state on the beginning, that: … The results also indicate an adverse effect of elevated leukocytosis on the first day of stroke, and of elevated CRP, on the risk of ICB and the functional status of patients on the first days following onset… . However, I could not find data concerning CRP in the results. Has CRP been studied?

  • Similarly, further authors widely discuss: … a correlation between the parameters of the inflammatory condition and functional status during the acute stroke period (platelets adherence, CD40 ligand and β- globulin, IL-6, platelet ratio (RPR), monocyte to high-density lipoprotein cholesterol ratio (MHR) and neutrophil to lymphocyte ratio (NLR), WBC counts and CRP levels etc… , however only impact of WBC on functional outcome has been Thus, in my opinion this prat of discussion should be substantially modified and better adapted to obtained results.

  • Authors also discuss … an adverse effect of elevated leukocytosis on the first day of stroke, and of elevated CRP, on the risk of ICB and the functional status of patients on the first days following onset. However, I can not find data regarding CRP impact on ICB among the results.

  • Study limitations are too sparse, in my opinion authors should declare and explain why the do not assess ASPECT data, especially they mention this widely used tool both, in introduction, and in the discussion.

Additionally, spelling, grammar and editing imperfections exists (ex. as follows), thus manuscript re-editing and language corrections would help, ex.:

  • their functional status according to modiefied Rankin scale (mRS) on days 10, 30 and 365 following stroke,

  • out Stroke Centre …

  • Table 4. … phenodata on mRS day 365…. Table 3. …. clinical phenodata on mRS day 30…. (do data really concerns experimental phenotypes ?)

  • A total of 417 stroke patients (mean age 67.8 ± 13.2 years; mean 70 [20–92]; female: 47.7%) hospitalized between 2019 and 2020, who were treated with mechanical thrombectomy due to stroke qualified for the retrospective study.

Author Response

Dear Reviewer,

We were happy to read the Reviewer’s and Editor's comments which we do appreciate. We have done our best to make sure that the paper itself, the changes introduced as per the Reviewer’s comments, and our responses meet your expectations.

Below we refer to each of the Reviewer’s comments and suggestions. The changes introduced to the text are highlighted in red.

Q & A:

Ad. Methods:

  1. Clear statement what are the study aims would improve data interpretation. First, authors declare that: … study aimed to identify the key parameters for efficacy of MT in stroke-patients in short and long-term follow-up treated with MT. However, further they state that … the key parameter for obtaining TICI 2b–3 could not be identified. In results, they generally present and conclude factors for the functional status in short and long-term observations. Thus, regarding presented material, in my opinion authors could simply declare, that they aimed to assess factors for functional outcome after MT in stroke patients.

A: We corrected some sentences in Introduction and Methods (aim of the study), according to valuable the Reviewer’s suggestion.

  1. Authors declare that they analyzed selected clinical and non-clinical factors, however they do not present selection criteria. Thus, Authors should present selection criteria, to avoid suspicion for “salami slicing” policy.

A: Thank you for above remark. We defined „clinical/non-clinical” parameters in section Methods and Results.

  1. In my opinion, too little is told about the selection criteria for variables involved into regression models. Description: …. To evaluate the accuracy of model predictions, the “leave-one-out” stoorę and the multiclass AUC estimator were used…. Is probably too sparse. AUC results are neither presented in the results.

 A: Thank you for above remark. We changed the description of statistic metods in section Methods.

Ad. Results 

Some imprecisions and unclarities exist, that should be modified for better quality of the manuscript.

  1. First, both in text and table 1 & 2 authors present some parameters, being described as … key parameters (in text - … An abnormal WBC count and the neurological status (NIHSS) in the ultra-acute phase of stroke were key parameters associated with ICB) or as those with the strongest effect (Tables). It remains unclear if The Authors mean that other variables has no statistically significant effect or impacted less? However, both other covariables and the significance of its impact should be presented or authors should prove why they have not been involved into the regression models.

A: We deeply changed the presentation of our results, to make them more clear and understandable. We remodeled and inserted the new tables with the clinical and non-clinical parameters and the results of the statistic analysis.  We included data (in tables and text) according to the comments and suggestions of the Reviewer.

  1. Term lipid disorders (Table 3) is oo general and should be clarified.

       A: we defined the lipid disorders (in section Methods)

Ad. Discussion

Some contradictions exist, that should be avoided, as follows:

  1. The Authors state on the beginning, that: … The results also indicate an adverse effect of elevated leukocytosis on the first day of stroke, and of elevated CRP, on the risk of ICB and the functional status of patients on the first days following onset… . However, I could not find data concerning CRP in the results. Has CRP been studied?

   A: Thank you for this remark. Perhaps, we have described this issue in little detail. Yes, we assessed the influence of CRP and leucocytosis on the neurological state and functional status. In modified text (Methods and Results) we inserted data concerning CRP and leucocytosis.

  1. Similarly, further authors widely discuss: … a correlation between the parameters of the inflammatory condition and functional status during the acute stroke period (platelets adherence, CD40 ligand and β- globulin, IL-6, platelet ratio (RPR), monocyte to high-density lipoprotein cholesterol ratio (MHR) and neutrophil to lymphocyte ratio (NLR), WBC counts and CRP levels etc… , however only impact of WBC on functional outcome has been Thus, in my opinion this prat of discussion should be substantially modified and better adapted to obtained results.

 A: We deleted some sentences from Discussion (sentences not directly relevant to our study)  according to the suggestions of the Reviewer.

  1. Authors also discuss … an adverse effect of elevated leukocytosis on the first day of stroke, and of elevated CRP, on the risk of ICB and the functional status of patients on the first days following onset. However, I can not find data regarding CRP impact on ICB among the results.

 A: Thank you for this remark. In the Discussion, we presented the results/observations the other authors. We corrected the paragraphs making this notation more unambiguous and understandable.  

  1. Study limitations are too sparse, in my opinion authors should declare and explain why the do not assess ASPECT data, especially they mention this widely used tool both, in introduction, and in the discussion.

A: We modified the Limitations according the Reviewer's suggestions.

  1. Additionally, spelling, grammar and editing imperfections exists (ex. as follows), thus manuscript re-editing and language corrections would help, ex.:
  2. their functional status according to modiefied Rankin scale (mRS) on days 10, 30 and 365 following stroke,

 A: We corrected the indicated sentence.

  1. out Stroke Centre …

A: We corrected the indicated sentence.

  1. Table 4. … phenodata on mRS day 365…. Table 3. …. clinical phenodata on mRS day 30…. (do data really concerns experimental phenotypes ?)

A: We corrected the indicated sentence.

  1. A total of 417 stroke patients (mean age 67.8 ± 13.2 years; mean 70 [20–92]; female: 47.7%) hospitalized between 2019 and 2020, who were treated with mechanical thrombectomy due to stroke qualified for the retrospective study.

A: We corrected the indicated sentence.

Reviewer 2 Report

The aim of this retrospective study was of investigating predictors of short and long-term outcome in patients with acute ischemic stroke undergoing mechanical thrombectomy (MT). After enrolling 417 patients, the authors observed  that age, NIHSS, diabetes, TICI score and first pass effect influenced mRS 10 days after stroke, while parameters with the strongest effect on the functional status on day 360 were: age and NIHSS on the first day of stroke. Although short-term results were largely expected, long-term data increased the originality of this study. However, the present version of this study needs to be improved.

Major comments:

  • The authors should clarify when NIHSS data were collected. What means “the first day of stroke”? At baseline? After the MT?
  • Since NIHSS and mRS are ordinal variables, report them as median and interquartile range.
  • Regarding laboratory parameters, the authors reported results only for white blood cell counts, C-reactive protein and troponin concentration. Although several other important parameters (e.g. hemoglobin, albumin) should be collected, I consider essential information on glucose levels. In fact, in addition to baseline glucose, recent studies showed that stress hyperglycemia may have a relevant role as outcome predictor in patients undergoing MT. I suggest the authors to dedicate a paragraph in their Discussion section on glucose/stress hyperglycemia and MT. Improve the reference list adding the studies of: Wang et al. J Stroke Cerebrovasc Dis 2019; Chen et al. J Stroke Cerebrovasc Dis 2019; Merlino et al. Front Neurol 2021.
  • The authors should report in the text in which manner vascular risk factors were diagnosed.
  • Information on stroke etiology based on the TOAST classification is due.
  • Authors should distinguished between symptomatic and asymptomatic hemorrhagic transformations.
  • Primary endpoints 90-days after stroke should be reported along with data at 360-days. I consider absolutely useless mRS data collected 10-days after stroke. Please delete them.

Author Response

Dear Reviewer,

we were happy to read the Reviewer’s and Editor's comments which we do appreciate. We have done our best to make sure that the paper itself, the changes introduced as per the Reviewer’s comments, and our responses meet your expectations.

Below we refer to each of the Reviewer’s comments and suggestions. The changes introduced to the text are highlighted in red.

Q & A:

  1. The authors should clarify when NIHSS data were collected. What means “the first day of stroke”? At baseline? After the MT?

A: We introduced the information about the time of NIHSS examination (at baseline, before thrombectomy) into the Methods section.

  1. Since NIHSS and mRS are ordinal variables, report them as median and interquartile range.

A: According to the Reviewers’ suggestion we introduced the median value and interquartile range (in Table 1).

  1. Regarding laboratory parameters, the authors reported results only for white blood cell counts, C-reactive protein and troponin concentration. Although several other important parameters (e.g. hemoglobin, albumin) should be collected, I consider essential information on glucose levels. In fact, in addition to baseline glucose, recent studies showed that stress hyperglycemia may have a relevant role as outcome predictor in patients undergoing MT. I suggest the authors to dedicate a paragraph in their Discussion section on glucose/stress hyperglycemia and MT. Improve the reference list adding the studies of: Wang et al. J Stroke Cerebrovasc Dis 2019; Chen et al. J Stroke Cerebrovasc Dis 2019; Merlino et al. Front Neurol 2021.

A: Thank you for the above suggestion. We have also the other parameters, which we have introduced to Table 1. We supplemented the Discussion with an additional paragraph on the interesting issue of stress hyperglycemia, we read the indicated literature sources and cited indicated authors.

  1. The authors should report in the text in which manner vascular risk factors were diagnosed.

A: The description of our diagnostic methods concerning the vascular risk factors in patients was inserted in the Methods section.

  1. Information on stroke etiology based on the TOAST classification is due.

A: We introduced the information about stroke etiology according to the TOAST classification (Results section).

  1. Authors should distinguished between symptomatic and asymptomatic hemorrhagic transformations.

A: We defined the symptomatic and asymptomatic intracranial bleeding and corrected paragraph in two sections- Methods and Results. We introduced a information about the symptomatic intracranial bleeding to the Table 1.

  1. Primary endpoints 90-days after stroke should be reported along with data at 360-days. I consider absolutely useless mRS data collected 10-days after stroke. Please delete them.

A: We deeply changed the description of our results and presentation them in the article. The table No 4 was deleted. Because the main goal of our research was to identify the key parameters for both short and long time after stroke, we maintained the result concerning the acute period of stroke (it means 10 day) The piece of information about the parameters influencing the functional status after MT we had to remain according to the suggestions of the second Reviewer. We hope the Reviewer will accept our modifications. We tried to cope with all suggestions/remarks of all Reviewers hoping improve our manuscript.

Round 2

Reviewer 1 Report

Authors modified their munuscript substantially and responded my remarks.

Reviewer 2 Report

The authors satisfied my requests.